# Recent Advance and Modification Strategies of Transition Metal Dichalcogenides (TMDs) in Aqueous Zinc Ion Batteries

**DOI:** 10.3390/ma15072654

**Published:** 2022-04-04

**Authors:** Tao Li, Haixin Li, Jingchen Yuan, Yong Xia, Yuejun Liu, Aokui Sun

**Affiliations:** 1School of Packaging and Materials Engineering, Hunan University of Technology, Zhuzhou 412007, China; 17404200515@stu.hut.edu.cn (T.L.); hxli_2002@163.com (H.L.); jcyuan_66@163.com (J.Y.); xiayong@hut.edu.cn (Y.X.); yjliu_2005@126.com (Y.L.); 2School of Metallurgy and Environment, Central South University, Changsha 410083, China

**Keywords:** transition metal dichalcogenides, aqueous zinc ion batteries, modification strategy, cathode materials

## Abstract

In recent years, aqueous zinc ion batteries (ZIBs) have attracted much attention due to their high safety, low cost, and environmental friendliness. Owing to the unique layered structure and more desirable layer spacing, transition metal dichalcogenide (TMD) materials are considered as the comparatively ideal cathode material of ZIBs which facilitate the intercalation/ deintercalation of hydrated Zn^2+^ between layers. However, some disadvantages limit their widespread application, such as low conductivity, low reversible capacity, and rapid capacity decline. In order to improve the electrochemical properties of TMDs, the corresponding modification methods for each TMDs material can be designed from the following modification strategies: defect engineering, intercalation engineering, hybrid engineering, phase engineering, and in-situ electrochemical oxidation. This paper summarizes the research progress of TMDs as cathode materials for ZIBs in recent years, discusses and compares the electrochemical properties of TMD materials, and classifies and introduces the modification methods of MoS_2_ and VS_2_. Meanwhile, the corresponding modification scheme is proposed to solve the problem of rapid capacity fading of WS_2_. Finally, the research prospect of other TMDs as cathodes for ZIBs is put forward.

## 1. Introduction

Due to the deterioration of the environment and the deficit of fossil energy, it is increasingly important to develop environmentally friendly, sustainable, and renewable energy [1,2,3]. At present, some renewable energy power generation can meet the requirements of environmental protection and sustainable development, such as solar power, wind power, and tidal power generation, but these energy sources suffer from regional limitations and instability, limiting their wider application. As an important part of energy for sustainable development, electrochemical energy storage has become an active research field in recent decades [4,5,6]. In the current energy market, lithium-ion batteries (LIBs) are dominant in automobile, medical equipment, portable wearable equipment, and other industries because of their excellent energy density and good environmental performance [7,8]. However, most lithium-ion batteries use organic solvents as electrolytes, which may cause safety problems and increase costs [9,10,11,12]. Compared with non-aqueous batteries, aqueous rechargeable batteries have the characteristics of low cost, non-toxicity, and non-flammability, which makes them safer, more environmentally friendly, and more economical [13,14,15].

In addition to LIBs, many other rechargeable aqueous metal ion batteries have been developed, including sodium ion batteries [16], potassium ion batteries [17], aluminum ion batteries [18], calcium ion batteries [19], and zinc ion batteries. Compared with other rechargeable aqueous metal ion batteries, zinc ion batteries have many advantages [20,21,22]: (1) Zinc ion batteries can be directly assembled in the air without inert environment, which can reduce the battery assembly cost. (2) Zinc ions can be electrodeposited reversibly in aqueous solution, so the zinc sheet can be directly used as the anode of the battery. (3) Zinc as anode has a higher theoretical capacity and lower oxidation/reduction potential (–0.76 V) than the standard hydrogen electrode, which indicates that there is a higher open circuit voltage when coupled with cathode. Therefore, in many rechargeable water-based metal ion batteries, the research on aqueous zinc ion batteries is increasingly concerned.

As shown in Figure 1a, the aqueous zinc ion battery (ZIB) is mainly composed of the battery shell, cathode material, anode material, electrolyte, and diaphragm. There are many kinds of cathode materials, including manganese-based materials [23], vanadium-based materials [24], and Prussian blue analogues [25], all of which possess a certain capacity of Zn^2+^ storage. Since the theoretical interlayer spacing of transition metal disulfide compounds is larger than the diameter of Zn^2+^, Zn^2+^ can be intercalated and deintercalated in TMD material, which also indicates that TMD material is feasible as the cathode of ZIB [26]. At present, few studies have been conducted on TMD as a cathode material of ZIB, and there is not a systematic exposition, which also shows from the aspect that TMD as cathode of ZIB is quite novel. Figure 1b displays the trend of the increasing number of publications, evidencing the increasing attention paid to TMD materials in ZIB. The latest research progress of TMD series materials as cathodes of ZIB in recent years is reviewed in this paper, and several modification methods which can improve the Zn^2+^ storage capacity and structural stability of TMD materials are expounded.

## 2. Characterization and Synthetic Methods of TMD

Two-dimensional layered transition metal disulfides (TMDs) generally have X-M-X structure, where M is the transition metal, such as molybdenum, vanadium, tungsten, bismuth, and other metal elements, while X is generally sulfur, selenium, and other elements, as shown in Figure 2a. These layered structures facilitate the transport of various carriers and can also adapt to the volume change during ion insertion. Because of their different chemical composition and unique crystal structure, as well as the fact that the d-orbitals can be filled with different elements, TMDs can be used as functional materials for electronic insulation, semiconductors, and superconductivity [27].

Among all TMDs materials, molybdenum disulfide (MoS_2_) has received extensive attention as a typical representative. Molybdenum disulfide is composed of two layers of sulfur atoms and one layer of molybdenum atoms, with the metal molybdenum layer interposed between the two sulfur layers, alternately stacked to form a sandwich-like structure. The sulfur atoms are bound together by van der Waals force, while the S-Mo-S atoms are linked by strong covalent bonds [28,29]. MoS_2_ not only has a layered structure, but also has different phases (1T, 2H, and 3R), and each phase also has different physical properties and chemical characteristics [30]. As shown in Figure 2b, MoS_2_ in 2H and 3R phases both demonstrate the triangular prismatic coordination of Mo atoms, and 2H-MoS_2_ is very stable because of the two layers of units stacked in hexagon symmetry. Meanwhile, 3R-MoS_2_ has rhombus symmetry, and each unit has three layers. On the contrary, the Mo atoms of 1T-MoS_2_ (metal phase) are octahedral coordinated and most unstable [31,32]. VS_2_ is also a common material in TMDs, where the sulfur and vanadium layers are stacked together in a sandwich-like structure by van der Waals force interactions [33,34]. Due to the large interlayer spacing, VS_2_ has great potential in the intercalation/deintercalation of ions, such as Li^+^, Na^+^, Zn^2+^, Mg^2+^, and Al^3+^ [35,36,37].

**Figure 2 materials-15-02654-f002:**
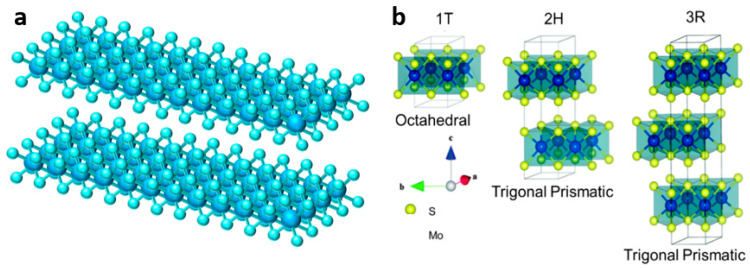
(**a**) Illustration of TMD structure; (**b**) Atomic structures of 1T-, 2H-, and 3R-MoS_2_ [31].

In order to obtain layered and nanoflower-like sulfide materials, TMD materials are generally prepared by hydrothermal and solvothermal methods. The synthetic methods of TMD materials as ZIB cathode are summarized in Table 1.

## 3. TMDs as ZIBs Cathode

Layered MoS_2_, VS_2_, WS_2_, and VSe_2_ have been proven to be feasible cathode materials for ZIB in recent years. MoS_2_ is the earliest and most extensively material studied in ZIB owing to its unique layered structure and different phases. However, compared with manganese-based materials and vanadium-based materials, TMDs have lower conductivity and larger capacity changes, which will lead to the lower rate performance and poor cycle performance of electrode materials during charge and discharge, which is also the biggest challenge for TMDs materials. In order to improve the electrochemical performance of TMDs, different modification strategies of TMDs were designed. In this section, the research progress of MoS_2_, VS_2_, WS_2_, and VSe_2_ materials as cathode materials for ZIB is described, and various modification methods of MoS_2_ and VS_2_ materials are discussed and summarized.

### 3.1. MoS_2_ and Modification 

Liu et al. [53] did not obtain obvious redox peaks in CV curves of MoS_2_ and WS_2_, indicating that it is difficult for MoS_2_ and WS_2_ to store zinc ions. The measured cyclic discharge capacities provided by MoS_2_ and WS_2_ were 18 mAh·g^−1^ and 22 mAh·g^−1^, respectively, and the EDS images of MoS_2_ and WS_2_ in the fully discharged state also had low zinc contents, so their conclusions showed that MoS_2_ and WS_2_ materials were not suitable as cathode materials for ZIB. The electrochemical performance of unmodified MoS_2_ in ZIB is undesirable, such that more and more scholars have paid attention to the modified MoS_2_ as the cathode material of ZIB.

#### 3.1.1. Defect Engineering

Defect engineering is used to change the surface properties and structure of TMDs to improve the electrochemical performance, which is undoubtedly a preferable modification strategy. The possible vacancy defects in TMDs consist of sulfur vacancies, transition metal vacancies, edge vacancies, and holes in the crystal lattice [54,55,56]. The main effects of defects on electrode materials are divided into three types: (1) More storage and adsorption sites of foreign ions are provided to improve the battery capacity. The introduction of defects increases the number of active sites significantly, which improves the electrochemical performance. (2) It changes the local structure and charge distribution on the surface of electrode materials, which can improve the conductivity, cycle stability, and rate performance of electrode materials. (3) Defects can make the structure more flexible and stable when external ions are intercalated and deintercalated. Therefore, the introduction of defects in the crystal structure of TMDs and the selection of appropriate chemical modification molecules or groups using the defect sites constructed on the surface of TMDs can improve the electrical conductivity, reversible capacity, and initial Coulombic efficiency (ICE) and enhance the reversibility of Zn^2+^ intercalation/de intercalation, thus improving the electrochemical performance of TMD electrode materials in ZIB.

For example, Xu et al. [38] designed the first MoS_2−x_/Zn cell and confirmed that defect engineering could be used to enhance the Zn storage capacity of MoS_2_. They predicted that the edge vacancies and sulfur vacancies were the main adsorption sites of Zn^2+^ by DFT, while the original surface was inert in storing Zn^2+^ due to the lack of these vacancies. As shown in Figure 3, owing to the absence of some chalcogen atoms, there is a large number of vacancies existing in the neatly arranged sulfur layer. These vacancies include edge vacancies and sulfur vacancies, which can easily accommodate more Zn^2+^ for intercalation, resulting in a significant increase in the reversible capacity of ZIBs. Such a layered nanostructure with rich defects can also reduce the diffusion energy barrier and significantly increase the diffusion rate of Zn ions [57]. Figure 4a shows that the capacity of defect-free MoS_2_ nanosheets is very low at a current density of 100 mA·g^−1^, while the defect-rich MoS_2_ nanosheets provide a high reversible capacity of 135 mAh·g^−1^. After 1000 cycles at a current density of 1000 mA·g^−1^, the defect-rich MoS_2_ electrode still delivers a high reversible capacity of 88.6 mAh·g^−1^ with a capacity retention of 87.8%. The electrochemical reaction mechanism of the MoS_2−x_/Zn battery at cathode and anode can be expressed as:Cathode:  MoS_2−x_ + 0.36Zn^2+^ + 0.72e^−^ → Zn_0.36_MoS_2−x_(1)
Anode:  Zn^2+^ + 2e^−^ → Zn(2)

#### 3.1.2. Interlayer Engineering

The non-bonded characteristics of 2D layered materials with unique vdW gaps between adjacent atomic layers allows the insertion of atoms, ions, and molecules without breaking covalent bonds through a chemical intercalation process, thus changing the physical and chemical properties of the material [58]. TMDs are generally composed of 2D single-molecule layers combined with van der Waals forces and weak chemical bonds stacked along the direction perpendicular to the layers, a structure that has a larger specific surface area and is also conducive to the insertion of small intercalators, such as ions and molecules. Figure 5 shows that the insertion of external ions can expand the interlayer spacing of TMDs.

Li et al. [39] prepared expanded interlayer distance E-MoS_2_ nanosheets vertically aligned on carbon fiber cloth by a one-step glucose-assisted hydrothermal method as cathode materials for ZIB. The layer spacing of commercial MoS_2_ is 0.62 nm, while the interlayer spacing of E-MoS_2_ nanosheets is 0.70 nm. The increased interlayer spacing is due to the incorporation of hydrated Na^+^ and NH_4_^+^ into the MoS_2_ crystal framework during the hydrothermal reaction. The E-MoS_2_ electrode also showed good performance with a specific capacity of 202.6 mAh·g^−1^ and a specific energy density of 148.2 Wh·kg^−1^ at 0.1 A·g^−1^ as well as a capacity retention rate of 98.6% after more than 600 cycles with good cycling stability. The electrochemical reaction mechanism of the cathode and anode of E-MoS_2_/Zn battery is expressed as:
Cathode:  xZn^2+^ + 2xe^−^+ MoS_2_ → Zn_x_MoS_2_(3)Anode:  xZn^2+^ + 2xe^−^ → xZn(4)

Liang et al. [40] made MoS_2_-O to store Zn ions more efficiently by adjusting the interlayer spacing and designing hydrophilicity tuning engineering. A small amount of oxygen (5%) was added to MoS_2_, and the incorporation of oxygen improved the hydrophilicity of MoS_2_-O and made it possible for water to intercalate in MoS_2_-O, which also increased the interlayer spacing of MoS_2_-O from 0.62 to 0.95 nm, as shown in Figure 4b. Furthermore, if a diffusing hydrated Zn^2+^ were to diffuse through a narrow void separation in MoS_2_ (S to S layer separation), i.e., 0.31 nm, instead of the 0.62 nm, i.e., Mo to Mo layer separation, interlayer engineering becomes even more crucial. As a result, the incorporation of oxygen increased the diffusion rate of Zn^2+^ by three orders of magnitude and the storage capacity of Zn^2+^ was increased by10 times. When the current density is 100 mA·g^−1^, the capacity reaches 232 mAh·g^−1^, which is higher than that of MoS_2_-based electrode materials in other ZIB.

Unusually, Zhang et al. [41] designed a material with crystalline water introduced into MoS_2_ nanoflowers as cathode materials for ZIB. The MoS_2_·nH_2_O nanoflowers obtained by the one-step hydrothermal method are composed of layered nanosheets with low stacking height and uniform distribution, as shown in Figure 4c. The insertion of crystalline water not only increases the interlayer distance of MoS_2_ to 0.65 nm, but also shields the electrostatic interaction between zinc ions and matrix materials, and lubricates and increases the diffusion ion channels as a way to promote the migration of Zn^2+^. Under the synergistic effect of crystal water lubrication and nanoflower structure, the reversible capacity of MoS_2_·nH_2_O was 165 mAh·g^−1^ at 0.1 A·g^−1^. At the high current density of 2 A·g^−1^, the capacity retention rate after 800 cycles was 88%. 

Huang et al. [42] first designed a molybdenum disulfide/polyaniline (MoS_2_/PANI) hybrid material with overlapping heterostructures as an excellent cathode for ZIB. The MoS_2_/PANI prepared by the hydrothermal method is a micro-nanoflower assembled by some ultra-thin wrinkled nanosheets with a layer spacing of 1.03 nm, as shown in Figure 4d. An appropriate amount of PANI was inserted into the MoS_2_ matrix, thus expanding the interlayer spacing and shielding the electrostatic interaction between some Zn^2+^ and MoS_2_ substrates. The intercalation of polyaniline not only expands the Zn^2+^ diffusion channel, but also reduces the band gap between the conduction band and the valence band of MoS_2_, which improves the charge transfer efficiency. Compared with the original MoS_2_, the zinc storage performance of the MoS_2_/PANI hybrid cathode was significantly improved. At a current density of 1.0 A·g^−1^, it provides a high reversible capacity of 106.5 mAh·g^−1^ and maintains 86% capacity after 1000 cycles. This also provides a modification strategy for developing other TMD materials as good cathode materials for ZIB.

#### 3.1.3. Hybridization

The conductivity of TMDs is poor, which limits the diffusion power of ions. In electrochemical energy storage, carbon nanotubes, carbon fiber cloth, and graphene are the most commonly used hybrid materials. Carbon nanotubes and graphene materials have unique structures, high electrical conductivity, good mechanical properties, and large specific surface areas [59,60]. As shown in Figure 6, coating with a certain material or hybridizing with carbon nanotubes, carbon cloth, and graphene on the surface of the TMD can increase the surface area and charge transfer ability of the material to improve the electrochemical performance of the composite.

Huang [43] dispersed multi-walled carbon nanotubes in aqueous solution mixed with glucose and deionized water, and prepared ultra-thin MoS_2_ nanosheets with expanded layer spacing directly grown on the surface of carbon nanotubes (CNTs) by a one-step hydrothermal method (as shown in Figure 4e) as the ZIB cathode material. When the interlayer spacing of MoS_2_ increases to 1.0 nm, the increase of interlayer spacing reduces the diffusion energy barrier of Zn^2+^ between layers, and the main chain of CNTs promotes the electron transfer of Zn^2+^ between layers. More importantly, the layered structure hinders the aggregation of MoS_2_ nanosheets, increases the surface area, and provides a large number of active sites for electrochemical reactions. At the current density of 0.1 A·g^−1^, the reversible capacity of the MoS_2_@CNTs electrode was about 180.0 mAh·g^−1^. When the current density increases to 3 A·g^−1^, it can maintain 51.7% capacity, and after 500 cycles (the current density is 1.0 A·g^−1^) it can maintain 80.1% capacity. Therefore, MoS_2_@CNTs as cathode can provide high reversible capacity and good cycle stability for ZIB.

#### 3.1.4. Phase Engineering

Phase engineering has been proven to be an effective method to regulate the electrochemical performance of TMDs [61]. According to different stacking methods, the materials can be divided into three bulk phases: 1T, 2H and 3R. The cell of 1T phase is octahedral, and the cell of 2H phase is prismatic. Due to different atomic configurations, single-layer TMD materials can also exist in three different forms of 1T, 2H, and 3R, respectively. MoS_2_ is a typical TMD which has different phase structures, different metal coordination geometries, different stacking sequences, and different properties. For example, the natural bulk MoS_2_ crystal has a 2H phase with triangular prism coordination geometry, but the coordination of molybdenum and sulfur atoms is octahedral in 1T phase which shows metallicity [62,63]. In fact, the conductivity of 1T phase MoS_2_ is about 105 times higher than that of the 2H phase MoS_2_ corresponding to semiconductor [64,65]. It can be reasonably speculated that MoS_2_ with different phases exhibits different properties in application. Phase engineering has also been proven to be an effective strategy to regulate the catalytic activity of MoS_2_ [66].

For example, Liu et al. [44] prepared MoS_2_ with different phases by regulating the reaction temperature of the hydrothermal reaction, and most samples showed a flower-like structure of self-assembled nanosheets. As the synthesis temperature decreases, the interlayer spacing of MoS_2_ increases from 0.62 nm to 0.68 nm, as shown in Figure 4f. Different from MoS_2_ nanosheets corresponding to 2H, MoS_2_ nanosheets with high 1T phase content (about 70%) have excellent specific capacity, and the reason for this is that the Zn^2+^ diffusion barrier of 1T-MoS_2_ is much lower than that of 2H-MoS_2_. At a current density of 1.0 A·g^−1^, the reversible capacity of 1T-MoS_2_ is about 140 mAh·g^−1^, but the capacity decreases rapidly after 100 cycles. However, the capacity attenuation of 2H-MoS_2_ can be ignored after 400 cycles, showing good cycle performance. How to make MoS_2_ show excellent specific capacity and maintain stable cycle performance is still worth exploring.

### 3.2. VS_2_ and Modification 

In recent years, vanadium disulfide (VS_2_) based nanomaterials have become potential electrode materials for various energy storage batteries due to their highly controllable structure and chemical composition. The VS_2_ layer shows metallicity and has a considerable electronic state at the Fermi level, which is conducive to promoting electron and electrochemical activity [67]. At present, using VS_2_ as the cathode of ZIB is still considered to be a relatively new method. So far, there are only a few reports on VS_2_ as the cathode of ZIB. For example, He et al. [45] firstly synthesized rose-like VS_2_ nanoflowers assembled by nanosheets with diameter of 5–8 μm and thickness of 50–100 nm, whose interlayer spacing is 0.576 nm, as shown in Figure 7a. These nanosheets can realize the intercalation/deintercalation of Zn^2+^ in VS_2_ nanosheets. At a current density of 0.05 A·g^−1^, VS_2_ nanosheets have a high capacity of 190.3 mAh·g^−1^ and an energy density of 123 Wh·kg^−1^. The capacity retention rate reached 98.0% after 200 cycles at a current density of 0.5 A·g^−1^, which confirmed the feasibility of VS_2_ as the cathode of ZIB. They summarize the electrochemical reactions occurring in the cathode of Zn/VS_2_ batteries, as shown in Equation (5):VS_2_ + 0.09Zn^2+^ + 0.18e^−^ → Zn_0.09_VS_2_; Zn_0.09_VS_2_ + 0.14Zn^2+^ + 0.28e^−^ → Zn_0.23_VS_2_(5)

In other reports, VS_2_ was modified to obtain electrode materials with better electrochemical performance. Consistently, we reviewed the research progress of modified VS_2_ as electrode material for ZIB and summarized two corresponding modification strategies in this section.

#### 3.2.1. Hybridization

Jiao et al. [46] synthesized hierarchical 1T-VS_2_ directly on stainless steel mesh (VS_2_@SS) by the hydrothermal method. The prepared VS_2_ was composed of bending nanosheets with a transverse size of 5–8 μm, forming a highly layered network flower structure from Figure 7b. There is a large number of interlayer channels for electrolyte penetration in VS_2_ nanoflowers, which enhances the contact area between electrolyte and VS_2_ and promotes the transmission of electrons and ions. Therefore, the VS_2_@SS electrode with VS_2_ loading of 4–5 mg·cm^−2^ has a high capacity of 198 mAh·g^−1^ at a current density of 50 mA·g^−1^. After 2000 cycles at a current density of 2 A·g^−1^, the capacity retention rate was about 80%, showing a superior cycle life. When the mass load of VS_2_ is about 11 mg·cm^−2^, the electrode can maintain 90% capacity in 600 cycles (only 0.017% loss per cycle), which indicates that the stability of the composite structure is enhanced after the hybrid of TMD material and SS.

Graphene and reduced graphene oxide (rGO) are idealized cathode materials with large specific surface area, excellent electrical conductivity, and excellent mechanical elasticity [68]. In addition, the end-functional group surface of rGO further provides the possibility of strong interface coupling between the active material and the conductive skeleton, and it also realizes the rapid charge transfer and high structural stability against repeated electrochemical cycles [69,70]. Due to the unique layered structure and large specific surface area of graphene and VS_2_, the electrochemical performance of the electrode can be further improved by hybridizing graphene and VS_2_. For example, Chen et al. [65] synthesized vertically grown ultra-thin VS_2_ nanosheets (rGO-VS_2_) on graphene sheets by the solvothermal method. The interlayer distance of these ultrathin VS_2_ nanosheets is about 0.97 nm, as shown in Figure 7c, which is much larger than that of commercial VS_2_ crystals (0.575 nm). They also calculated that the surface area of rGO-VS_2_ was 34.2 m^2^·g^−1^, which was larger than that of commercial VS_2_ (26.5 m^2^·g^−1^), indicating that the introduction of graphene increased the specific surface area of the composites. VS_2_ nanosheets are closely connected with graphene, which can effectively prevent the dissolution of VS_2_ and improve the cycle stability of the battery. Therefore, when the current density is 0.1 A·g^−1^, it has a high capacity of 238 mAh·g^−1^. At a high current density of 5 A g^−1^, the capacity is 190 mAh·g^−1^, and the capacity retention is over 93% after 1000 cycles.

Metal oxides, polymers, C_3_N_4_, and MXene can also be hybridized with TMD materials [71,72,73]. Pu et al. [48] prepared a rose-like VS2@VOOH material with hydrophilic VOOH coating for the first time. The size of VS_2_@VOOH is about 10 μm and a large number of nanosheets are assembled into uniform rose-like morphology, as shown in Figure 7d. Hydrophilic VOOH coating is conducive to the penetration of electrolyte in ZIB, and the structure is still intact even after O-H is exchanged with water in VOOH. Their research also showed that the O-H bond in VOOH could not only improve the permeation of electrolyte to the electrode, but also reduce the possibility of vanadium dissolution. Therefore, after 350 cycles at a current density of 1.5 A·g^−1^, the specific capacity of VS_2_@VOOH composite can reach 107.5 mAh·g^−1^. Even after 400 cycles at a current density of 2.5 A·g^−1^, the VS_2_@VOOH material still maintained a specific capacity of 91.4 mAh·g^−1^, which further verified the potential of the modified VS_2_ as a ZIB cathode.

#### 3.2.2. In Suit Electrochemical Oxidation

Electrochemical oxidation, as a feasible and multi-purpose strategy, has been proven to enhance the electrochemical behavior of low-valent active substances relative to evolutionary forms and newly formed high-active substances [74,75,76,77]. As shown in Figure 8, electrochemical oxidation of the inserted ions into the corresponding oxides without changing the position of the ion insertion into the TMD can provide more active sites or enhance the hydrophilicity of the ions, which is beneficial to the diffusion of Zn^2+^ and the expansion of the TMD layer spacing. Yang et al. [49] prepared a hollow spherical VS_2_·NH_3_ material with layer spacing expansion as the cathode of ZIB. The VS_2_·NH_3_ hollow flower ball assembled by nanosheets has a porous structure, and the expanded layer spacing is 0.98 nm, as shown in Figure 7e. They found that during the first charge process, the expanded VS_2_·NH_3_ transformed into V_2_O_5_·nH_2_O with a layer spacing of 1.21 nm after electrochemical oxidation. Then, Zn^2+^ was intercalated/deintercalated in V_2_O_5_·nH_2_O with large interlayer spacing, which provided high capacity and good stability for subsequent charge–discharge cycles. When the current density is 0.1, 0.2, 0.5, 1.0, 2.0, and 5.0 A·g^−1^, the capacity of the Zn/VS_2_·NH_3_ battery is 390, 372, 320, 260, 189, and 107 mAh·g^−1^, respectively, which is much higher than the VS_2_-based cathode of other ZIB previously reported. It is worth noting that the electrode formed by in-situ electrochemical oxidation into V_2_O_5_·nH_2_O reached an amazing 110% capacity retention compared with the initial capacity after 2000 cycles at a high current density of 3 A·g^−1^.

Yu et al. [50] first used the in situ electrochemical pretreatment of VS_2_ in aqueous medium and obtained VS_2_/VO_x_ heterostructures as cathode materials for ZIB. The fluffy VO_x_ nanosheets are uniformly grown on VS_2_, forming an interwoven porous electrode rich in VS_2_ and VO_x_ heterostructures. The VS_2_/VO_x_ heterostructure combines the high conductivity of VS_2_ and the high chemical stability of VO_x_, which is conducive to regulating the intercalation of guest ions and improving the chemical stability of the VS_2_ skeleton. The step-by-step insertion of Zn^2+^ changed the buffer volume, making the reaction kinetics faster and more reversible. As shown in Figure 7f, the VS_2_/VO_x_ electrode maintains 75% high capacity in 3000 cycles at a current density of 1 A·g^−1^. After 1000 cycles, VS_2_/VO_x_ still maintained the original crystal structure and nanosheet morphology, which again proved that the VS_2_/VO_x_ heterostructure had high reversibility. As a consequence, the design of TMD heterostructures by in situ electrochemical oxidation provides a new strategy for the efficient energy storage of ZIB.

### 3.3. WS_2_ and Modification 

WS_2_ is a natural metal sulfide, where the S atom is located in the lattice position of the closely packed hexagonal structure. The plane where the W atom lies is sandwiched between two S layers and each W atom is coordinated with the S atom. The W layer and the S layer are stacked together through a weak van der Waals effect to form a prismatic structure [78,79]. The theoretical layer spacing of WS_2_ is 0.618 nm, while the diameter of Zn^2+^ is only 0.404~0.43 nm. Theoretically, it can also be used as the electrode material of the energy storage battery, but there are few reports on it as the ZIB cathode material. A report [53] on WS_2_ pointed out that the interlayer spacing of commercial WS_2_ was insufficient, and the inherent conductivity of Zn^2+^ for reversible intercalation between layers was low, which indicates that the storage capacity of commercial WS_2_ for Zn^2+^ was poor. Another reason for this is that 2H-WS_2_ has a large Zn^2+^ intercalation barrier, which seriously hinders the diffusion of Zn^2+^ to electrode materials.

Tang et al. [51] proved for the first time that 1T-WS_2_ nanosheets are a promising candidate cathode material for ZIB. Thus, 1T phase WS-200 is stacked by ultrathin nanosheets, and the interlayer spacing is expanded from 0.618 nm to about 0.90 nm. After the introduction of 1T-WS_2_, the capacity of an almost inactive Zn/WS_2_ battery increased by eight times, and the reversible capacity was 179.99 mAh·g^−1^ at a current density of 200 mA·g^−1^. However, WS_2_ has the problem of rapid capacity degradation in the long cycle at different current densities from Figure 9a. Their explanation for this phenomenon is that the layered WS_2_ structure collapsed and dissolved, and some 1T-WS_2_ dissolved in the electrolyte during the intercalation/deintercalation of Zn^2+^, which provides a research direction for improving the reversible capacity and cycle stability of WS_2_.

In order to improve the electrochemical performance of WS_2_, a modification strategy similar to MoS_2_ and VS_2_ can be designed to modify WS_2_, the most important of which is hybridization and intercalation. For example, Debela et al. [80] synthesized WS_2_@NGr material by hybridizing WS_2_ with nitrogen-doped graphite. The reversible capacity of WS_2_@NGr electrode reached 963 mAh·g^−1^. Ratha et al. [81] synthesized layered WS_2_ and reduced graphene oxide (RGO) hybrid materials by a simple hydrothermal method. WS_2_/RGO composites showed excellent cycle stability in supercapacitors. Liu et al. [82] successfully synthesized a highly stable 1T-WS_2_ nanobelt with a special zigzag chain superlattice structure. The NH_4_^+^ insertion not only extends the interlayer spacing of WS_2_ to 0.95 nm, but also makes the structure highly stable. Although the above three WS_2_ modified materials are not applied to ZIB, they also provide many strategies. Starting from the microstructure of WS_2_ and the storage mechanism of Zn^2+^ in WS_2_, the carbon-based materials and WS_2_ can be hybridized, or external ions, such as NH_4_^+^ and crystal water, can be used for intercalation, so as to overcome the challenge of rapid capacity decline faced by WS_2_.

### 3.4. Other Materials

VSe_2_ is also a typical layered structure, and the vanadium layer is alternately located between two selenium layers, forming a sandwich structure formed by van der Waals interaction [83,84]. Larger interlayer spacing (0.611 nm) provides sufficient transport channels and active sites for active cations. Moreover, the electronic coupling force between adjacent V^4+^ -V^4+^ induces metal properties and excellent electronic conductivity, making it an attractive candidate electrode material [85,86]. Wu et al. [52] prepared ultra-thin VSe_2_ nanosheets and used them as cathode materials for ZIB. As shown in Figure 9b, the specific capacities of VSe_2_ are 250.6 and 132.6 mAh·g^−1^ at 200 and 5000 mA·g^−1^, respectively. After 800 cycles at 2 A·g^−1^, the capacity retention was 83%. The factors influencing the excellent electrochemical performance of VSe_2_ nanosheets cathode are attributed as follows: (1) Highly reversible Zn^2+^ intercalation/deintercalation storage mechanism; (2) Thanks to its ultra-thin two-dimensional morphology, Zn^2+^ has rapid diffusion kinetics (DZn^2+^ ≈ 10^−8^ cm^−2^ s^−1^); (3) The metal characteristics of VSe_2_ are conducive to the thermodynamics and kinetics of Zn^2+^ storage process; (4) Stability of crystal structure of VSe_2_ electrode material during long cycle. They summarized the storage mechanism of VSe_2_/Zn battery as 6 and 7:Cathode:  VSe_2_ + 0.23Zn^2+^ + 0.46e^−^ → Zn_0.23_VSe_2_Zn_0.23_VSe_2_ + 0.17Zn^2+^ + 0.34e^−^ → Zn_0.4_VSe_2_(6)Anode:  0.4Zn^2+^ + 0.8e^−^ → 0.4Zn(7)


A few other TMD materials can also be used in the water-based zinc ion battery. For example, Zhang et al. [87] proved that CoS_2_ could be used as anode material for a sulfur redox type non-aqueous zinc battery. Li et al. [88] synthesized a novel pre-cured titanium disulfide (Na_0.14_TiS_2_) as anode material for ZIB. However, these materials are all anode materials for ZIB, which are not described too much here.

## 4. Conclusions and Future Work

Since ZIB offer good safety and environmental protection, and with the deepening research on ZIB, they are expected to become a substitute for LIB. The special layered structure of TMDs is conducive to the transport of carriers such as Zn^2+^. The nanoflower-like structure synthesized by the ordered stacking of monolayers confers TMD materials with a larger specific surface area. Some TMDs also have interlayer spacing larger than the diameter of hydrated zinc ion. Therefore, TMD materials are considered to represent a candidate cathode of ZIB with significant potential. Although TMDs are conductive, they also face the challenges of low reversible capacity and rapid capacity decay.

In recent years, some studies have reported that the modified TMD materials have excellent reversible capacity and cycle stability as cathodes of ZIB. Based on these research reports, the research progress of typical TMDs materials, such as MoS_2_, VS_2_, WS_2_, and VSe_2_, as cathodes of ZIB was described in this paper. The reversible capacity and long-period cycle performance are summarized in Table 2 and compared. At the same time, the modification ideas of TMD cathode materials for ZIB were combed, and the corresponding modification methods were summarized and discussed: (1) defect engineering, providing more ion storage sites and active sites by constructing chemical defects; (2) interlayer engineering, namely through the intercalation of guest ions to expand the layer spacing; (3) hybridization with carbon-based materials or other materials, making the microstructure of the composites more stable or maintained in a stable phase structure; (4) phase engineering, controlling the mutual transformation between 1H and 2R phases to reduce the diffusion energy barrier of Zn^2+^ and make the structure more stable; (5) in-situ electrochemical oxidation, without changing the position of ions in the interlayer, where the low-valent active substances are oxidized into kinetic high-valent active substances by electrochemistry.

Although these modification strategies are currently only applied to MoS_2_ and VS_2_, they should also be applicable to other TMD materials because the TMDs have similar special layered structures. For example, in view of the rapid decline of WS_2_ capacity, we may venture to assume that the hybridization of WS_2_ and carbon-based materials improves the structural stability of the composites, thus improving the poor cycle stability of WS_2_ electrodes. With the modification strategies put forward, TMD materials are bound to receive more attention, and more series members of TMDs will become excellent cathode materials for ZIB.

## Figures and Tables

**Figure 1 materials-15-02654-f001:**
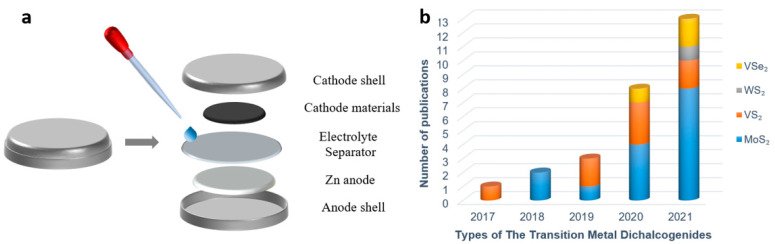
(**a**) Disassembly diagram of coin-type ZIBs; (**b**) The trend of publications on TMDs as ZIBs cathode materials.

**Figure 3 materials-15-02654-f003:**
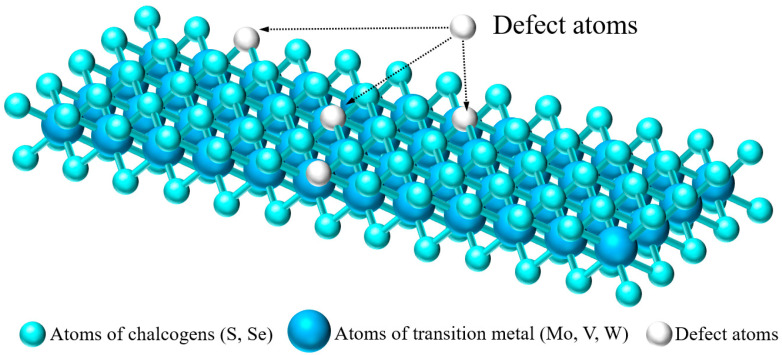
Defect atoms in TMDs singer layer.

**Figure 4 materials-15-02654-f004:**
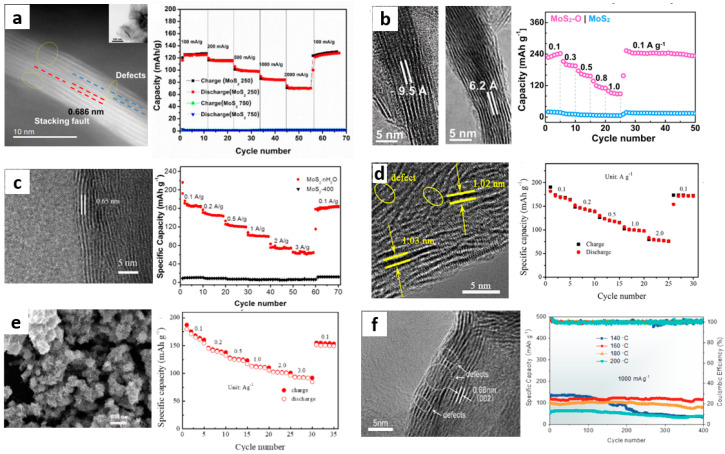
(**a**) HRTEM image of the defect−rich MoS_2_ and its rate performance [38]; (**b**) HRTEM images of oxygen incorporated MoS_2_ (left showing oxygen incorporated MoS_2_, right showing pristine MoS_2_) and the rate performance of oxygen incorporated MoS_2_ [40]; (**c**) HRTEM image of MoS_2_·nH_2_O and the rate performance of Zn/MoS_2_·nH_2_O batteries [41]; (**d**) HRTEM image of the MoS_2_/PANI−150 hybrid and its rate performance [42]; (**e**) SEM image and rate performance of the MoS_2_@CNTs hybrid electrode [43]; (**f**) HRTEM image of MoS_2_−160, and cycling performances at current densities of 1.0 A g^−1^ of MoS_2_ at various temperatures [44].

**Figure 5 materials-15-02654-f005:**
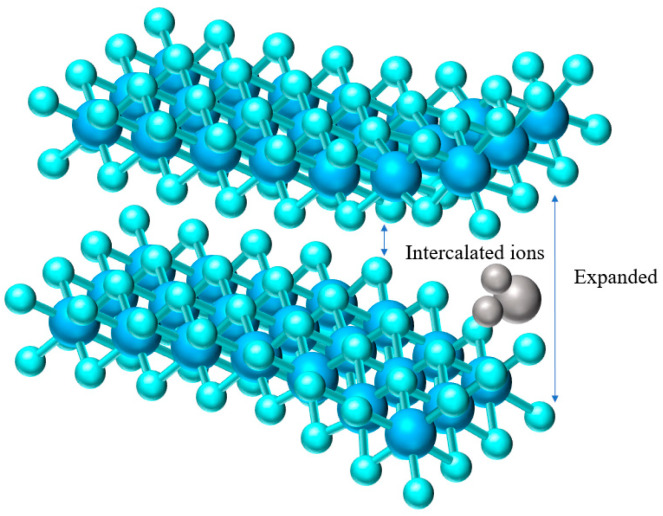
Ion Intercalation in single layer TMDs structure.

**Figure 6 materials-15-02654-f006:**
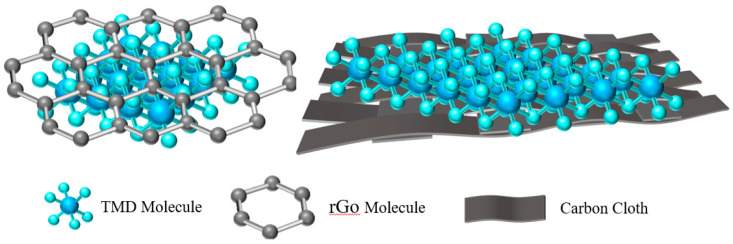
Modification of TMDs by Hybrid Engineering.

**Figure 7 materials-15-02654-f007:**
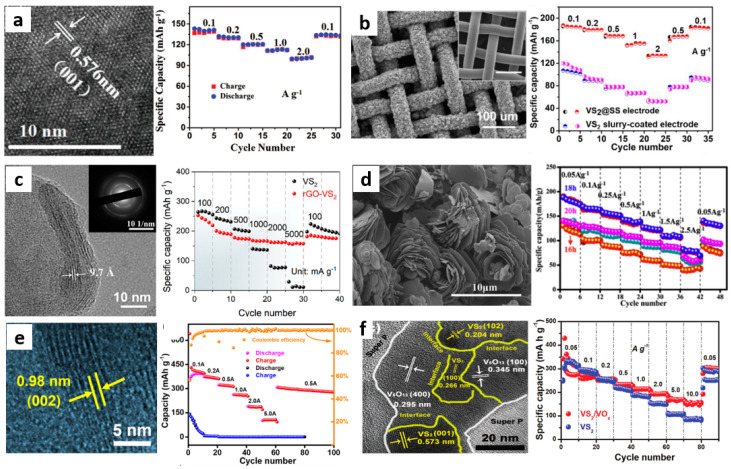
(**a**) HR−TEM image and rate performance of layered VS_2_ [45]; (**b**) SEM image of VS_2_ grown on SS mesh and rate performance of VS_2_@SS electrode [46]; (**c**) HRTEM image and rate performance of the rGO−VS_2_ composites [47]; (**d**) SEM image and rate performance of the VS_2_@VOOH−18h [48]; (**e**) HRTEM image and rate performance of VS_2_·NH_3_ electrode [49]; (**f**) TEM image and rate capability of in−situ electrochemical oxidation formed VS_2_/VO_x_ [50].

**Figure 8 materials-15-02654-f008:**
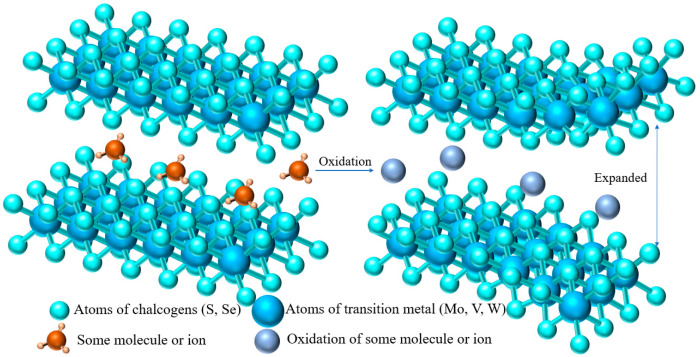
Modification of TMDs by in-situ electrochemical oxidation.

**Figure 9 materials-15-02654-f009:**
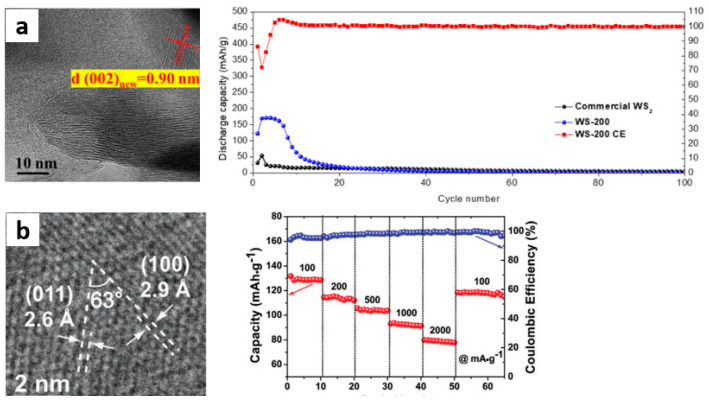
(**a**) HRTEM image, Cycling performances at current density of 200 mA g−^1^ of WS−200 [51]; (**b**) HRTEM image and rate performance of VSe_2_ [52].

**Table 1 materials-15-02654-t001:** Fabrication methods, precursors and synthesis conditions of TMDs as ZIBs cathode.

Products	Method	Precursors	Temperature	Duration	Ref
MoS_2−x_	Hydrothermal	(NH_4_)_6_Mo_7_O_24_·4H_2_O, TAA	200 °C	18 h	[38]
E- MoS_2_	Hydrothermal	Na_2_MoO_4_, CS(NH_2_)_2_, carbon cloth, glucose, HCl	190 °C	24 h	[39]
MoS_2_-O	Hydrothermal	(NH_4_)_6_Mo_7_O_24_·4H_2_O, thiourea	180 °C	24 h	[40]
MoS_2_·nH_2_O	Hydrothermal	(NH_4_)_6_Mo_7_O_24_·4H_2_O, thiourea	170 °C	24 h	[41]
MoS_2_/PANI	Solvothermal	Na_2_MoO_4_, thiourea, C_17_H_33_CO_2_Na, ethanol, OA, HCl	180 °C	24 h	[42]
MoS_2_@CNTs	Hydrothermal	Na_2_MoO_4_·2H_2_O, thiourea, CNTs, glucose	200 °C	24 h	[43]
MoS_2_-160	Hydrothermal	(NH_4_)_6_Mo_7_O_24_·4H_2_O, thiourea	160 °C	24 h	[44]
VS_2_	Hydrothermal	NH_4_VO_3_, TAA, NH_3_·H_2_O	180 °C	20 h	[45]
VS_2_@SS	Hydrothermal	NH_4_VO_3_, TAA, NH_3_·H_2_O, stainless steel mesh	180 °C	10 h	[46]
rGO-VS_2_	Solvothermal	VO(acac)_2_, cysteine, GO, NMP	200 °C	8 h	[47]
VS_2_@VOOH	Hydrothermal	V_2_O_5_, TAA, NH_3_·H_2_O	180 °C	18 h	[48]
VS_2_·NH_3_	Solvothermal	VO(acac)_2_, TAA, NMP	200 °C	24 h	[49]
VS_2_/VO_x_	Solvothermal	Na_3_VO_4_·12H_2_O, TAA, ethylene	180 °C	20 h	[50]
1T-WS_2_	Solvothermal	WCl6, TAA, DMF	200 °C	24 h	[51]
VSe_2_	Chemical Liquid Phase Synthesis	VO(acac)_2_, Se powder, OAm	330 °C	5 h	[52]

**Table 2 materials-15-02654-t002:** Summary of the electrochemical performance of TMDs as ZIBs cathode.

Cathode Material	Electrolyte	Voltage	Capacity [mAh•g^−1^]	Cycle Stability	Ref
MoS_2_-O	3M Zn (CF_3_SO_3_)_2_	0.2–1.4V	232 at 0.1 A g^−1^	68% after 2000 cycles at 1.0 A g^−1^	[40]
E- MoS_2_	2M Zn (CF_3_SO_3_)_2_	0.3–1.5V	202.6 at 0.1 A g^−1^	98.6% after 600 cycles at 1.0 A g^−1^	[39]
MoS_2_/PANI	3M Zn (CF_3_SO_3_)_2_	0.2–1.3V	181.6 at 0.1 A g^−1^	86% after 1000 cycles at 1.0 A g^−1^	[42]
MoS_2_@CNTs	3M Zn (CF_3_SO_3_)_2_	0.3–1.2V	180 at 0.1 A g^−1^	80.1% after 500 cycles at 1.0 A g^−1^	[43]
MoS_2_-160	3M Zn (CF_3_SO_3_)_2_	0.25–1.25V	168 at 0.1 A g^−1^	98.1% after 400 cycles at 1.0 A g^−1^	[44]
MoS_2_·nH_2_O	3M Zn (CF_3_SO_3_)_2_	0.2–1.25V	165 at 0.1 A g^−1^	88% after 800 cycles at 2.0 A g^−1^	[41]
MoS_2−x_	3M Zn (CF_3_SO_3_)_2_	0.25–1.25V	138.6 at 0.1 A g^−1^	87.8% after 1000 cycles at 1.0 A g^−1^	[38]
VS_2_·NH_3_	2M Zn (CF_3_SO_3_)_2_	0.2–1.7V	390 at 0.1 A g^−1^	110% after 2000 cycles at 3.0 A g^−1^	[49]
VS_2_/VO_x_	25M ZnCl_2_	0.1–1.8V	260 at 0.1 A g^−1^	75% after 3000 cycles at 1.0 A g^−1^	[50]
rGO-VS_2_	3M Zn (CF_3_SO_3_)_2_	0.4–1.7V	238 at 0.1 A g^−1^	93% after 1000 cycles at 5.0 A g^−1^	[47]
VS_2_@SS	1M ZnSO_4_	0.4–1.0V	187 at 0.1 A g^−1^	80% after 2000 cycles at 2.0 A g^−1^	[46]
VS_2_@VOOH	3M Zn (CF_3_SO_3_)_2_	0.4–1.0V	165 at 0.1 A g^−1^	86% after 200 cycles at 0.5 A g^−1^	[48]
VS_2_	1M ZnSO_4_	0.4–1.0V	159.1 at 0.1 A g^−1^	98% after 200 cycles at 0.5 A g^−1^	[45]
WS-200	1M ZnSO_4_	0.1–1.5V	206.25 at 0.1 A g^−1^	0% after 100 cycles at 0.2 A g^−1^	[51]
VSe_2_	2M ZnSO_4_	0.1–1.6V	131.8 at 0.1 A g^−1^	80.8% after 500 cycles at 0.1 A g^−1^	[52]

## Data Availability

Not applicable.

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
