# Peer review of "Recent Advance and Modification Strategies of Transition Metal Dichalcogenides (TMDs) in Aqueous Zinc Ion Batteries"

_materials, 2022, doi:10.3390/ma15072654_

Round 1

Reviewer 1 Report

The review is publishable following the suggested revisions described below.

Fig. 1b shows only 15 publications. Is this correct?

“Zinc ions can be electrodeposited reversibility”, reversibly

Fig. 1. Check Fig. 1a for possible mistakes in assignment of the various parts.

“delaminated”, deintercalated

“CV images”, images??

“The introduction of defects also increases the large number of active sites, promoting the reaction kinetics effectively and enhancing the electrochemical phase transition.”. This is poorly written.

Fig. 3 needs more labels and description in the text.

Equation (1): why not 0.72?

Eq. (3) and (4): 2x, not x2.

Fig. 6 is described before Fig. 5.

“This strategy broadens our horizon and holds great research prospects in effectively exploiting the intercalation of crystalline water to enhance the Zn2+ storage capacity of other TMD materials.” This is not needed, delete.

Fig. 5 is not very informative or scientifically accurate.

Eq. (6) is not needed.

“(more than commercial VS2)”, this is not clear, explain with some citation also.

Fig. 8 should contain labels explaining the various molecules and species.

“while the particle diameter of Zn2+”, particle??

“monolayer layered structure”, poor English

“zinc hydrate”, hydrated zinc ion

Table 1. Sort the rows in terms of descending capacity for ease of reading.

Author Response

Dear Editors and Reviewers:

Thank you for your letter and for the reviewers’ comments concerning our manuscript entitled “Recent Advance and Modification Strategies of Transition Metal Dichalcogenides (TMDs) in Aqueous Zinc Ion Batteries”. Those comments are all valuable and very helpful for revising and improving our paper, as well as the important guiding significance to our researches. We have studied comments carefully and have made correction which we hope meet with approval. Revised portion have been annotated in the manuscript. The main corrections in the paper and the responses to the reviewers’ comments are as follows:

Reviewer #1:

  1. Fig 1b shows only 15 publications. Is this correct?

Response: We double-checked the Web of Science for articles published before March 2022 and found that there were indeed omissions. In the fields of MoS2, VS2, WS2 and VSe2 as cathodes for ZIB, there are 15,8,1 and 3 articles, respectively. We list the details of them, as shown in the table below. The data in Fig 1b has also been updated.

MoS2

Pub Date

Journal

Volume

DOI

Title

1

2018.9.13

Energy Storage Materials

16

10.1016/j.ensm.2018.09.009

Defect Engineering Activating (Boosting) Zinc Storage Capacity of MoS2

2

2018.10.12

Energy Storage Materials

19

10.1016/j.ensm.2018.10.005

MoS2 Nanosheets with Expanded Interlayer Spacing for Rechargeable Aqueous Zn-Ion Batteries

3

2019.4.15

Nano Letters

19

10.1021/acs.nanolett.9b00697

Aqueous Zinc Ion Storage in MoS2 by Tuning the Intercalation Energy

4

2020.2.10

Chemical Engineering Journal

389

10.1016/j.cej.2020.124405

Boosting Aqueous Zinc-Ion Storage in MoS2 via Controllable Phase

5

2020.7.1

International Journal of Electrochemical Science

15

10.20964/2020.07.64

Synthesis and Properties of Halloysite Templated Tubular MoS2 as Cathode Material for Rechargeable Aqueous Zn-ion Batteries

6

2020.9.30

ChemElectroChem

7

10.1002/celc.202001036

Hierarchical MoS2@CNTs hybrid as a long life and high rate cathode for aqueous rechargeable Zn-ion batteries

7

2020.12.11

Energy Storage Materials

35

10.1016/j.ensm.2020.12.010

Boosting zinc-ion intercalation in hydrated MoS2 nanosheets toward substantially improved performance

8

2021.2.17

Advanced Materials

33

10.1002/adma.202007480

Sandwich-Like Heterostructures of MoS2/Graphene with Enlarged Interlayer Spacing and Enhanced Hydrophilicity as High-Performance Cathodes for Aqueous Zinc-Ion Batteries

9

2021.3.17

Journal of Alloys and Compounds

871

10.1016/j.jallcom.2021.159541

Toward fast zinc-ion storage of MoS2 by tunable pseudocapacitance

10

2021.3.23

Journal of Alloys and Compounds

872

10.1016/j.jallcom.2021.159599

Crystal water assisting MoS2 nanoflowers for reversible zinc storage

11

2021.5.10

Nanoscale Advances

3

10.1039/d1na00166c

A nano interlayer spacing and rich defect 1T-MoS2 as cathode for superior performance aqueous zinc-ion batteries

12

2021.5.18

Electrochimica Acta

388

10.1016/j.electacta.2021.138624

Tuning the kinetics of zinc ion in MoS2 by polyaniline intercalation

13

2021.7.14

ACS Applied Materials & Interfaces

13

10.1021/acsami.1c11063

Nitrogen-Doped Metallic MoS2 Derived from a Metal–Organic Framework for Aqueous Rechargeable Zinc-Ion Batteries

14

2021.11.5

Scripta Materialia

209

10.1016/j.scriptamat.2021.114368

Flexible design of large layer spacing V-MoS2@C cathode for high-energy zinc-ion battery storage

15

2021.11.23

Journal of Alloys and Compounds

898

10.1016/j.jallcom.2021.162854

Template-assisted hydrothermal synthesized hydrophilic spherical 1T-MoS2 with excellent zinc storage performance

VS2

1

2017.1.6

Advanced Energy Materials

7

10.1002/aenm.201601920

Layered VS2 Nanosheet-Based Aqueous Zn Ion Battery Cathode

2

2019.6.15

Journal of Materials Chemistry A

7

10.1039/c9ta04798k

Binder-Free Hierarchical VS2 Electrodes for High-Performance Aqueous Zn Ion Batteries towards Commercial Level Mass Loading

3

2019.7.22

Journal of Power Sources

437

10.1016/j.jpowsour.2019.226917

Rose-like vanadium disulfide coated by hydrophilic hydroxyvanadium oxide with improved electrochemical performance as cathode material for aqueous zinc-ion batteries

4

2020.8.19

Journal of Power Sources

477

10.1016/j.jpowsour.2020.228652

VS2 nanosheets vertically grown on graphene as high-performance cathodes for aqueous zinc-ion batteries

5

2020.12.23

Advanced Functional Materials

31

10.1002/adfm.202008743

Boosting Zn2+ and NH4+ Storage in Aqueous Media via In-Situ Electrochemical Induced VS2/VOx Heterostruct

6

2020.12.31

Applied Surface Science

544

10.1016/j.apsusc.2020.148882

A highly stable aqueous Zn/VS2 battery based on an intercalation reaction

7

2021.3.16

Journal of Materials Chemistry C

9

10.1039/d1tc00531f

A VS2@N-doped carbon hybrid with strong interfacial interaction for high-performance rechargeable aqueous Zn-ion batteries

8

2021.9.1

Journal of Colloid and Interface Science

607

10.1016/j.jcis.2021.08.194

Boosting the zinc ion storage capacity and cycling stability of interlayerexpanded vanadium disulfide through in-situ electrochemical oxidation strategy

WS2

1

2021.10.20

Journal of Alloys and Compounds

894

10.1016/j.jallcom.2021.162391

Investigation of zinc storage capacity of WS2 nanosheets for rechargeable aqueous Zn-ion batteries

VSe2

1

2020.8.9

Small

16

10.1002/smll.202000698

Ultrathin VSe2 Nanosheets with Fast Ion Diffusion and Robust Structural Stability for Rechargeable Zinc-Ion Battery Cathode

2

2021.5.10

ACS Applied Materials & Interfaces

13

10.1021/acsami.1c04596

Selenium Defect Boosted Electrochemical Performance of Binder-Free VSe2 Nanosheets for Aqueous Zinc-Ion Batteries

3

2021.6.8

Journal of Alloys and Compounds

882

10.1016/j.jallcom.2021.160704

Nanohybrid engineering of the vertically confined marigold structure of rGO-VSe2 as an advanced cathode material for aqueous zinc-ion battery

  1. “Zinc ions can be electrodeposited reversibility”, reversibly

Response: Considering the reviewer’s suggestion, we have modified improper English expressions.

  1. Fig.1. Check Fig. 1a for possible mistakes in assignment of the various parts.

Response: We have carefully checked the position of each part and found that the zinc anode and cathode materials were placed in the wrong place. We have amended it in the new figure.

  1. “delaminated”, deintercalated

Response: We are very sorry for our incorrect writing and it has been corrected in the text.

  1. “CV images”, images??

Response: We are very sorry for our incorrect writing and it has been corrected in the text. It should be “CV curves”.

  1. “The introduction of defects also increases the large number of active sites, promoting the reaction kinetics effectively and enhancing the electrochemical phase transition.”. This is poorly written.

Response: We are very sorry for our incorrect writing and it has been corrected as “The introduction of defects makes the number of active sites increased significantly, which improves the electrochemical performance.”.

  1. Fig. 3 needs more labels and description in the text.

Response: The labels of the Fig.3 and relevant description have been added.

  1. Equation (1): why not 0.72?

Response: In the literature 38 the data is 0.728, but we believe that 0.72 is the correct one. Therefore, in the text we changed to 0.72.

  1. Eq. (3) and (4): 2x, not x2.

Response: We have made the correction.

  1. Fig.6 is described before Fig.5.

Response: We have carefully checked the descriptions of all the figures in the article and rearranged their order.

  1. “This strategy broadens our horizon and holds great research prospects in effectively exploiting the intercalation of crystalline water to enhance the Zn2+ storage capacity of other TMD materials.” This is not needed, delete.

Response: We have deleted relevant words.

  1. Fig.5 is not very informative or scientifically accurate.

Response: TMDs are generally hybridized with carbon-based materials and other TMD materials. Therefore, we amended Fig. 5 to depict the schematic representation of the hybridization of the TMD material with rGo and carbon cloth (see Fig. 6 in revised manuscript).

  1. Eq. (6) is not needed.

Response: The Eq. (6) has been deleted in the text.

  1. “(more than commercial VS2)”, this is not clear, explain with some citation also.

Response: It refers to the average load of commercial VS2's on the stainless steel mesh. We have removed it to avoid any ambiguity.

  1. Fig. 8 should contain labels explaining the various molecules and species.

Response: The labels have been added in the Fig. 8.

  1. “while the particle diameter of Zn2+”, particle??

Response: We have deleted the word.

  1. “monolayer layered structure”, poor English

Response: We have modified relevant expressions.

  1. “zinc hydrate”, hydrated zinc ion

Response: We have made the correction.

  1. Table 1. Sort the rows in terms of descending for ease of reading.

Response: Thanks for the suggestions. We have arranged the contents of the Table in descending order (see Table 2 in revised manuscript).

Reviewer 2 Report

(1) The authors can include the general synthetic methods of TMDs. 

(2) The authors can include the general structure of chalcogenides for better understanding 

(3) The authors may refer the other batteries with same cathode materials and various intercalated metal ions (other than Zn2+) and comparision of their efficiency

*****

Author Response

Thank you for your letter and for the reviewers’ comments concerning our manuscript entitled “Recent Advance and Modification Strategies of Transition Metal Dichalcogenides (TMDs) in Aqueous Zinc Ion Batteries”. Those comments are all valuable and very helpful for revising and improving our paper, as well as the important guiding significance to our researches. We have studied comments carefully and have made correction which we hope meet with approval. Revised portion have been annotated in the manuscript. The main corrections in the paper and the responses to the reviewers’ comments are as follows:

  1. The authors can include the general synthetic methods of TMDs.

Response: Thank you for your suggestion. We have summarized the synthesis methods of the materials described in the manuscript, as shown in the Table 1 in revised manuscript.

  1. The authors can include the general structure of chalcogenides for better understanding.

Response: The general structure of TMD is shown in Fig 2(a).

  1. The authors may refer the other batteries with same cathode materials and various intercalated metal ions (other than Zn2+) and comparision of their efficiency.

Response: This review focuses on TMD materials as ZIB cathodes, and it would be a bit off-topic to compare TMD applications in other batteries. Moreover, relevant reviews have been published on the application and performance comparison of TMD in other batteries, and references are as follows:

  1. Sahoo R., Singh M., Rao T. N. A Review on the Current Progress and Challenges of 2D Layered Transition Metal Dichalcogenides as Li/Na-ion Battery Anodes. ChemElectroChem. 2021, 8, 2358-2396.
  2. Chang U., Eom K. Enhancing the Capacity and Stability of a Tungsten Disulfide Anode in a Lithium-Ion Battery Using Excess Sulfur. ACS Appl. Mater. Interfaces. 2021, 13, 20213–20221.
  3. Eunjeong Yang,Hyunjun Ji,Yousung Jung, Two-Dimensional Transition Metal Dichalcogenide Monolayers as Promising Sodium Ion Battery Anodes. J. Phys. Chem. C. 2015, 119, 26374–26380.

We tried our best to improve the manuscript and made some changes in the manuscript. These changes will not influence the content and framework of the paper. We appreciate for Editors/Reviewers’ warm work earnestly, and hope that the correction will meet with approval. Once again, thank you very much for your comments and suggestions.

Best regards,

Aokui Sun
